# Siberian wildfire smoke observations from space-based multi-angle imaging: A multi-year regional analysis of smoke particle properties, their evolution, and comparisons with North American boreal fire plumes

- 5 Katherine T. Junghenn Noyes<sup>1,2</sup> and Ralph A. Kahn<sup>3,4</sup>
  - <sup>1</sup> Earth System Science Interdisciplinary Center, University of Maryland, College Park MD, 20740
  - <sup>2</sup> Earth Sciences Division, NASA Goddard Space Flight Center, Greenbelt MD 20771
  - <sup>3</sup> The Laboratory for Atmospheric and Space Physics, University of Colorado Boulder, Boulder CO 80303
  - <sup>4</sup> Senior Research Scientist Emeritus, NASA Goddard Space Flight Center, Greenbelt MD 20771

Correspondence to: Katherine T. Junghenn Noyes (junghenn@umd.edu)

#### Abstract.

10

The physical and chemical properties of biomass burning (BB) smoke particles vary with fuel type and burning conditions, greatly affecting their impact on climate and air quality. However, the factors affecting smoke particle properties are not well characterized on a global scale, and the factors controlling their evolution during transport are not well constrained. From observations of the Multi-Angle Imaging Spectrometer (MISR) instrument aboard NASA's Terra satellite, smoke aerosol optical depth (AOD) can be retrieved, along with constraints on near-source plume vertical extent, smoke age, and particle size, shape, light-absorption, and absorption spectral dependence. Previous work demonstrated the robust, statistical characterization of BB particles across Canada and Alaska using MISR and other remote sensing data. Here we expand upon this work, studying over 3,600 wildfire plumes across Siberia. We leverage the MISR Research Aerosol (RA) algorithm to retrieve smoke particle properties and the MISR Interactive Explorer (MINX) tool to retrieve plume heights and the associated wind vectors. These results are compared statistically to available observations of fire radiative power (FRP), land cover characteristics, and meteorological information. Correlations appear between the retrieved smoke particle properties, smoke age, local ambient conditions, and fuel type, allowing us in many cases to infer the dominant aging mechanisms and the timescales over which they occur. Specifically, we find that plumes located in areas with higher peat content are subject to less oxidation and condensation/hydration compared to other plume types (e.g., forest and grassland), and are predominantly affected by dilution instead.

Deleted:

Deleted: d

## 1. Introduction

Wildfires are an integral component of the Earth system, influencing ecosystem processes across the globe. Although a certain degree of fire activity is natural and expected, the past two decades have been marked by a surge in large, uncontrolled fires that often take significant tolls on human society (e.g., Bowman et al., 2009). This phenomenon is only expected to worsen in the coming decades due to changing temperature and precipitation patterns brought on by man-made climate change (Liu et al., 2014; IPCC, 2021). The already fire-prone regions of eastern Australia, the Amazon, Canadian and Alaskan boreal zones, southern Europe, and the western United States may experience enhanced fire activity, and areas that were not previously particularly disposed to significant fire activity may become more fire-prone.

Wildfires have significant potential to impact regional air quality and climate conditions -- often emitting substantial amounts of trace gases and airborne particles, which can alter atmospheric chemistry and physics across time and space. The precise impacts of wildfire smoke depend upon plume transport processes, smoke composition, and particle evolution, which can vary widely. Although CO2 and water vapor tend to dominate emissions, smoke from wildfires is a rich mixture of greenhouse gases such as methane, volatile and semi-volatile organic gases/particles, light-scattering aerosols, and the lightabsorbing particles containing black carbon (BC) and brown carbon (BrC). Globally, wildfires are the most significant source of light-absorbing airborne particles (Bond et al., 2013; Feng et al., 2013). In addition to exhibiting distinct chemical properties, BC and BrC are optically unique in that BC is highly absorbing across all visible wavelengths, whereas BrC is less absorbing overall and displays enhanced light absorption at shorter wavelengths (Kirchstetter et al., 2004; Samset et al., 2018). These light-absorbing particles have the potential to affect the local radiative budget, impacting atmospheric stability (Taubman et al., 2004; Liu et al., 2014), whereas light-scattering smoke aerosols may seed clouds by serving as cloud-condensation nuclei (CCN) and thus increase cloud albedo (Albrecht, 1989; Kaufman and Fraser, 1997; Koch and Del Genio, 2010; Warner and Twomey, 1967; Hansen et al 1997; Hobbs and Radke, 1969). Often, smoke aerosols also contribute towards poor air quality regionally, as particulate matter is dangerous to respiratory health and can produce long-term medical consequences (Naeher et al., 2007; Reid et al., 2016; Elser et al., 2024), and lofted smoke plumes through plume-rise processes can lead to long-range horizontal transport. Such plume-rise processes may lead to smoke aerosols escaping the planetary boundary layer (PBL) and entering the free troposphere (FT; Kahn et al., 2008), where they can stay aloft for several days or more, travel great distances, and affect conditions far downwind (Damoah et al., 2004; Taubman et al., 2004; Vant-Hull et al., 2005; Colarco et al., 2004).

The microphysical properties and mixing state of smoke particles can change dramatically even a short distance away from the source. For example, particles may increasingly undergo oxidation as they mix with background air, trace gases, and sunlight, leading to both chemical and physical changes (Zhou et al., 2017; Kleinman et al., 2020). As smoke cools downwind, semi-volatile gases (known as volatile organic compounds, or VOCs) can condense onto existing emitted particles, creating organic or inorganic coatings that result in increased particle size and alter particle scattering and absorption as well as CCN efficiency, especially in the case of BC, which is hydrophobic in its pure form. These VOCs can also spontaneously condense into new, small particles (Wang et al., 2013; Yokelson et al., 2009; Akagi et al., 2012; Hennigan et al., 2012; Ahern et al.,

Deleted: However.

Deleted: ; Kahn et al., 2008; Liu et al., 2014

Deleted: away from the flame front

Deleted: (

Deleted: )

Deleted: Reid et al., 2005;

Deleted: Zhou et al., 2017:

2019; Kleinman et al., 2020; Hodshire et al., 2019). These and other processes often occur in combinations that may change on relatively short temporal and spatial scales. However, the factors that determine which mechanism or mechanisms affect the observable particle properties most are currently not well understood.

Wildfires display a range of fire behavior and smoke characteristics that depend on factors such as vegetation type and fuel structure, terrain characteristics, and soil, climate, and weather conditions; together, they influence, among other things, the relative degree of flaming or smoldering combustion at the source. Differences in smoke particle properties are at least partially linked to differences in fire regimes and environmental conditions, with evidence suggesting systematic differences in particle size distribution, particle light absorption, and the spectral dependence of absorption (Dubovik et al., 2002; Eck et al., 2003; Junghenn Noyes et al., 2022; Shi et al., 2019; O'Neill et al., 2002). For example, studies have suggested a connection between fire regime and particle size at the point of emission, with smoldering fires (lower combustion efficiency, or CE) generating larger particles than flaming fires (higher CE) under many conditions (Reid and Hobbs, 1998; Reid et al., 2005). These fire regimes have also been linked to smoke particle type – although BC is often the dominant absorbing aerosol component in biomass burning (BB) smoke, smoldering fires tend to produce higher fractions of BrC than flaming ones

(Chakrabarty et al., 2010, 2016; Petrenko et al., 2012).


Based on current knowledge of the factors controlling smoke particle properties, we might expect that geographic and meteorological conditions are important drivers of particle speciation, plume chemistry, and evolution. However, to date there have been no global observational studies to help constrain these relationships on a large scale. As wildfire frequency and severity are expected to increase, it is becoming increasingly important to improve our understanding of the factors controlling wildfire smoke particle properties and to constrain these parameters for climate modeling. In previous work, we leveraged the multi-angle, multi-spectral observing capabilities of the Multi-Angle Imaging Spectroradiometer (MISR) instrument aboard NASA's Terra satellite to study wildfire smoke across multiple years in Canada and Alaska (Junghenn Noyes et al., 2022). Using the MISR Interactive Explorer tool (Nelson et al., 2013) to retrieve plume heights and their associated wind vectors, as well as to estimate smoke age, and the MISR Research Aerosol (RA) algorithm (Limbacher et al., 2022) to qualitatively determine particle properties, it was shown that there are distinct patterns in particle light-absorption, size, and the timescales over which particles evolve, based on the dominant land cover type present in the fire.

In the current work, we apply the MISR tools to a large ensemble of smoke plumes across Siberia, expanding upon the methods used in our previous study of Canadian and Alaskan wildfires in Junghenn Noyes et al., 2022. This represents the second instalment of an ongoing effort to characterize wildfire smoke particles in different ecosystems across the globe, with the eventual goal of building a global, region-specific inventory that (1) characterizes emitted and evolved smoke particle properties; (2) identifies patterns and establishes relationships among fuel type, burning conditions, ambient meteorology, and plume properties; and (3) infers the relevant aging mechanisms and associated timescales from the observed patterns. To this end, we compare the retrieved patterns associated with different ecosystems and environmental conditions with an array of other data, including fire radiative power (FRP) and land cover type from the MODerate resolution Imaging Spectroradiometer

Deleted: ; Chen et al., 2008

**Deleted:** (Junghenn Noyes et al., 2022)

(MODIS), and meteorological reanalysis from the Modern Era Retrospective-analysis for Research and Applications (MERRA-2). Trends in particle properties are also studied in the context of smoke age estimates derived from MINX wind vectors. Statistical analysis of the relationships among these observations provides insight into the factors controlling BB particle type emissions and allows inferences about the associated aging processes, directly addressing key elements missing from current climate and air quality modeling efforts.




Siberia was chosen as the second region of study for our global inventory because of its distinct similarities and differences with our previous Canada and Alaska work, and the important role the different biomes play in the fire-climate system. Further, they represent the only two major boreal zones (Siberia, in a broad sense, spans across continental Russia from the Ural Mountains to the Pacific Ocean) in which major biomass burning frequently occurs. These regions also represent the largest forest biomes in the world, and are about one-third of global forest cover (de Groot et al., 2013). In the last several decades, northern-latitude boreal fires have released significant carbon into the atmosphere, which accelerates warming over the Arctic and leads to a positive feedback loop in warming (Lavoué et al., 2000). Wildfires are also the most important factor in taiga dynamics, as larch and Scots pine trees have evolved under conditions of periodic fires to gain an evolutionary advantage over less-adapted species; in permafrost zones such fires are necessary for larch to thrive (Kharuk et al., 2021). In addition, periodic wildfires decrease the danger of fuel buildup that can lead to larger, catastrophic fires having the potential to destroy natural ecosystems and encroach on human settlements if not managed.

Despite the important role boreal fires play in the Earth system, there is still a considerable lack of understanding of fire activity and variability in permafrost zones. In addition, the North American and Eurasian boreal zones have different vegetation and meteorological characteristics that translate into differences in wildfire characteristics and smoke properties, although these factors are currently not well characterized. For example, wildfires in Russia tend to be mostly limited to the surface, and therefore are usually of low-to-moderate in intensity, whereas wildfires in Canada are more likely to occur as high intensity crown fires (de Groot et al., 2013; Kharuk et al., 2021). From this, one might expect Siberian wildfires to emit larger fractions of BrC than North American boreal fires, but this has not yet been studied on a large scale. Extensive work has been done quantifying particle and trace gas emissions locally in boreal regions via field campaigns and ground-based remote studies (e.g. Wiggins et al., 2021; Junghenn Noyes and Kahn, 2024) and modeling (e.g., Soja et al., 2004; Kukavskaya et al., 2012). MISR offers a unique opportunity to observe both biomes on a large scale over multiple burning seasons.

Section 2 describes the data and methodology used in this study. Results and discussion are given in Sect. 3. Conclusions are presented in Sect. 4.

## 2. Methodology

## 2.1. The MISR Instrument

The MISR instrument flies aboard the Terra satellite, which resides in a polar orbit that crosses the equator at ~10:30 AM local time. With a ~380 km swath width, it provides global sampling about once per week, more frequently toward the poles. There are nine cameras aboard, viewing Earth in the nadir, forward, and aft directions along the orbital path (0°, +/-26.1°, +/-45.6°, +/-60.0°, and +/-70.0°), with four spectral bands observed at each angle, centered at approximately 446, 558, 672, and 866 nm (Diner et al., 1998). MISR's unique multi-angle, multi-spectral capabilities allow for: 1) the stereoscopic retrieval of height and motion vectors of contrast features in clouds and aerosol plumes, and 2) the radiometric retrieval of aerosol amount along with key aerosol optical properties that can be used to constrain particle type.

Table 1. Summary of the main features for the MISR components included in the Research Aerosol (RA) retrieval climatology, using the algorithm version summarized in Sect. 2.1. SSA558 denotes single-scattering albedo at 558 nm; r<sub>c</sub> denotes effective radius. Table S1 in the Supplement quantifies the particle refractive indices and SSA at each wavelength as well as the upper/lower bounds and variance for the different particle size distributions.

| Particle Shape <sup>a</sup> | Particle Size a,b         | Particle<br>Light-Absorption <sup>a</sup> | Particle Light-<br>Absorption Slope <sup>a</sup> | Analog Name       |
|-----------------------------|---------------------------|-------------------------------------------|--------------------------------------------------|-------------------|
|                             |                           | Strongly absorbing                        | Flat                                             | Black Smoke (BlS) |
|                             | Very Small                | (SSA <sub>558</sub> ~0.80)                | Steep                                            | Brown Smoke (BrS) |
|                             | (r <sub>e</sub> ~0.06 μm) | Moderately absorbing                      | Flat                                             | Black Smoke (BlS) |
|                             | (1ε0.00 μπ)               | (SSA <sub>558</sub> ~0.90)                | Steep                                            | Brown Smoke (BrS) |
|                             |                           | Non-absorbing                             | N/A                                              | Non-absorbing     |
|                             |                           | (SSA <sub>558</sub> =1.0)                 | 14/14                                            |                   |
|                             |                           | Strongly absorbing                        | Flat                                             | Black Smoke (BIS) |
|                             | Small                     | (SSA <sub>558</sub> ~0.80)                | Steep                                            | Brown Smoke (BrS) |
|                             | (r <sub>e</sub> ~0.12 μm) | Moderately absorbing                      | Flat                                             | Black Smoke (BIS) |
| Spherical                   | (16 -0.12 μm)             | (SSA <sub>558</sub> ~0.90)                | Steep                                            | Brown Smoke (BrS) |
| Spilerical                  |                           | Non-absorbing                             | N/A                                              | Non-absorbing     |
|                             |                           | (SSA <sub>558</sub> =1.0)                 | IV/A                                             |                   |
|                             | ·                         | Strongly absorbing                        | Flat                                             | Black Smoke (BIS) |
|                             | Medium                    | (SSA <sub>558</sub> ~0.80)                | Steep                                            | Brown Smoke (BrS) |
|                             | (r <sub>e</sub> ~0.26 μm) | Moderately absorbing                      | Flat                                             | Black Smoke (BlS) |
|                             | (160.20 μπ)               | (SSA <sub>558</sub> ~0.90)                | Steep                                            | Brown Smoke (BrS) |
|                             |                           | Non-absorbing                             | N/A                                              | Non-absorbing     |
|                             |                           | (SSA <sub>558</sub> =1.0)                 | IN/A                                             |                   |
|                             | Large                     | Non-absorbing                             | N/A                                              | Non-absorbing     |
|                             | $(r_e \sim 1.28~\mu m)$   | (SSA <sub>558</sub> =1.0)                 | IV/A                                             |                   |
| Non-spherical               | Large                     | Weakly absorbing                          | N/A                                              | Soil or dust      |
| 11011-spilerieai            | $(r_e \sim 1.21~\mu m)$   | (SSA <sub>558</sub> ~0.95)                | IV/A                                             |                   |

<sup>&</sup>lt;sup>a</sup> Particles are classified using four elements: shape — spherical vs. non-spherical; size — very small, small, medium, and large; light-absorption — non-absorbing, weakly absorbing, moderately absorbing, strongly absorbing; and spectral light-absorption profile — equal in all bands (flat) or varying between bands (steep). The SSA values at each wavelength can be found in Table S1 in the Supplement.

**Deleted:** The particle refractive indices and SSA values at each wavelength are provided in

b Each component size category assumes a lognormal distribution around a designated effective radius refurther details can be found in Table SI in the Supplement).





The MISR Interactive Explorer (MINX) software tool calculates plume heights and wind vectors by observing the parallax of contrast features within a plume (Nelson et al., 2008, 2013). MINX is a user-friendly, publicly available tool (https://github.com/nasa/MINX, last access: 21 March 2025) in which the operator manually identifies the plume source, plume extent, and wind direction in the MISR imagery, from which MINX determines heights and winds locally. In conditions where the plume exhibits sufficient contrast features and optical thickness relative to the surface, MINX plume height estimates are accurate to within +/- 0.5 km or better. MINX has been used extensively to retrieve heights and wind vectors for volcanic plumes (e.g., Kahn and Limbacher, 2012; Scollo et al., 2012; Flower and Kahn, 2017a, b, 2018, 2020a, b; Kahn et al., 2024) wildfire plumes (e.g., Val Martin et al., 2010, 2018; Tosca et al., 2011; Vernon et al., 2018; Junghenn Noyes et al., 2020a, b, 2022), and dust plumes (e.g., Yu et al., 2018; Kahn and Limbacher, 2025). For this work and related wildfire studies (e.g., Junghenn Noyes et al., 2022), we use the MINX wind vectors, along with the distance from the source in the MISR images, to calculate approximate plume-age horizons at discrete intervals within each plume snapshot. From this, we are then able to associate any patterns in the downwind evolution of smoke particles with general timescales. We also use MINX stereo heights to determine whether plumes were injected into the free troposphere, and to study how plume height and thickness may relate statistically to the retrieved smoke particle properties, ambient weather conditions, burning intensity as indicated by fire radiative power (FRP; assessed as the 4 µm brightness temperature anomaly), and fuel characteristics based on land cover type mapping. Approximately a third of the plumes included in this study were identified and digitized with MINX as part of the MISR Plume Height Project (Nelson et al., 2013), through which we obtained the data. The others were digitized specifically for the current study.

The MISR Research Aerosol (RA) algorithm compares the observed MISR radiances to a look-up table of simulated top-of-atmosphere (TOA) reflectances for a range of aerosol amounts and types, to retrieve aerosol optical depth (AOD), and to constrain particle extinction Ångström exponent (ANG), particle shape (spherical vs. non-spherical), and particle single-scattering albedo (SSA) and its spectral slope. The algorithm climatology can be customized depending on the intended use of the RA (e.g., wildfire vs. volcanic vs. dust plumes), with each particle "type" having distinct size and light-absorption properties. The retrieved particle property information is qualitative, typically with three to five bins in size (e.g., "small," "medium," or "large"), two to four bins in SSA (e.g., "highly absorbing," "weakly absorbing," and non-absorbing), and spherical vs. non-spherical in shape (Kahn et al., 2010; Kahn and Gaitley, 2015). For light-absorbing aerosols, particle type is further classified by the variation in SSA across the four MISR wavelengths, where the light-absorption by "flat" aerosols displays little to no wavelength dependence and is modelled on the spectral properties of urban pollution or black smoke (BIS), whereas the light-absorption by "steep" aerosols decreases with wavelength and is more typical of that observed in brown smoke (BrS) (Chen et al., 2008; Samset et al., 2018; Limbacher and Kahn, 2014; Andreae and Gelencser, 2006).

Formatted: Subscript

Deleted:

Deleted: defines

Formatted: Indent: First line: 0.5"

The algorithm determines the most likely mixture of aerosol candidates for each ~1.1-km MISR pixel, interpreted as the fraction of total mid-visible aerosol optical depth that can be attributed to each particle type. For wildfire studies, the algorithm climatology is currently comprised of one non-spherical optical analog and 16 spherical components ranging in size and SSA. Spherical particles are categorized as very small, small, medium, or large in size, strongly absorbing, weakly absorbing, or non-absorbing in the mid-visible range, and spectrally flat or spectrally steep in SSA. The non-spherical component is a large and weakly absorbing dust analog. Table 1 summarizes the RA component climatology, and more detailed information can be found in Table S1 in the Supplement. The operation of the RA is described by Limbacher and Kahn (2014, 2019; Limbacher et al., 2022). It should be noted that for high-confidence MISR aerosol-type retrievals, AOD usually needs to exceed 0.5, depending on the surface properties, and a given component needs to comprise at least 20% of the total AOD (Kahn et al., 2001; Kahn and Gaitley, 2015). As such, near-source smoke plumes generally provide excellent signal/noise for MISR particle property retrievals.

Table 2. Basic plume properties for all Siberian plumes included in this study, stratified by month across all years 2017-2021.


|                             |        | Apr.     | May      | Jun.     | Jul.     | Aug.     | Sept.    | All Mos. |
|-----------------------------|--------|----------|----------|----------|----------|----------|----------|----------|
| No. Plumes                  |        | 117      | 112      | 539      | 1600     | 1165     | 183      | 3716     |
| Plumes Above                | No.    | 15       | 19       | 110      | 168      | 103      | 1        | 416      |
| 2 km AGL                    | (%)    | (12.82%) | (16.96%) | (20.41%) | (10.5%)  | (8.84%)  | (0.55%)  | (11.19%) |
| Plumes Above the            | No.    | 22       | 36       | 184      | 407      | 341      | 58       | 1048     |
| MERRA-2 PBL                 | (%).   | (18.8%)  | (32.14%) | (34.14%) | (25.44%) | (29.27%) | (31.69%) | (28.20%) |
| 100 W 11                    | mean   | 1.240    | 1.422    | 1.530    | 1.360    | 1.335    | 0.826    | 1.349    |
| MINX Median<br>Plume Height | +/- σ  | 0.736    | 0.567    | 0.631    | 0.531    | 0.494    | 0.364    | 0.555    |
| riume rieignt               | median | 1.076    | 1.410    | 1.448    | 1.301    | 1.267    | 0.737    | 1.281    |
| AFFEDD 1 A                  | mean   | 1.524    | 1.552    | 1.652    | 1.524    | 1.527    | 0.906    | 1.514    |
| MERRA-2<br>PBL Height       | +/- σ  | 0.395    | 0.399    | 0.487    | 0.404    | 0.424    | 0.348    | 0.445    |
| I DL Height                 | median | 1.497    | 1.525    | 1.628    | 1.508    | 1.486    | 0.874    | 1.490    |
| MERRA-2                     | mean   | 6.041    | 4.204    | 3.738    | 5.661    | 4.774    | 5.038    | 5.041    |
| PBL-Top Stability           | +/- σ  | 5.164    | 3.656    | 3.137    | 3.505    | 2.832    | 3.441    | 3.397    |
| TBE-Top Stability           | median | 3.959    | 3.596    | 2.868    | 4.872    | 4.375    | 4.339    | 4.327    |
| MODIS Median                | mean   | 32.529   | 33.542   | 31.269   | 33.747   | 32.757   | 28.464   | 32.773   |
| MODIS Median<br>Plume FRP   | +/- σ  | 18.737   | 20.913   | 34.089   | 28.273   | 28.874   | 22.744   | 28.731   |
| Tiulic TKI                  | median | 28.100   | 28.675   | 25.800   | 26.525   | 25.000   | 22.250   | 25.900   |
| MICD M. P.                  | mean   | 0.724    | 1.026    | 0.995    | 1.438    | 1.606    | 1.215    | 1.380    |
| MISR Median<br>Plume AOD    | +/- σ  | 0.338    | 0.654    | 0.609    | 1.071    | 1.154    | 0.898    | 1.039    |
| Tiulic AOD                  | median | 0.673    | 0.820    | 0.848    | 1.155    | 1.284    | 0.980    | 1.086    |
| MERRA-2                     | mean   | 0.677    | 0.635    | 0.673    | 0.593    | 0.527    | 0.491    | 0.583    |
| 24-Hour Soil Moisture       | +/- σ  | 0.112    | 0.110    | 0.139    | 0.134    | 0.105    | 0.065    | 0.134    |
| 24-110ar Son Moistare       | median | 0.700    | 0.623    | 0.682    | 0.554    | 0.497    | 0.494    | 0.545    |
|                             | mean   | 0.876    | 0.886    | 0.924    | 0.940    | 0.931    | 0.873    | 0.928    |
| MISR SSA <sub>558</sub>     | +/- σ  | 0.037    | 0.043    | 0.040    | 0.034    | 0.036    | 0.046    | 0.041    |
|                             | median | 0.873    | 0.883    | 0.928    | 0.948    | 0.941    | 0.878    | 0.938    |
| MISR ANG                    | mean   | 1.867    | 1.750    | 1.713    | 1.648    | 1.624    | 1.878    | 1.671    |
| MISK ANG                    | +/- σ  | 0.245    | 0.270    | 0.269    | 0.274    | 0.251    | 0.239    | 0.272    |

|                                       |        | _     |       |       |       |       |       |       |
|---------------------------------------|--------|-------|-------|-------|-------|-------|-------|-------|
|                                       | median | 1.927 | 1.836 | 1.757 | 1.639 | 1.625 | 1.921 | 1.679 |
| Man No                                | mean   | 0.687 | 0.643 | 0.523 | 0.437 | 0.519 | 0.771 | 0.506 |
| MISR BIS<br>AOD Fraction              | +/- σ  | 0.237 | 0.270 | 0.272 | 0.247 | 0.220 | 0.215 | 0.256 |
| AOD Fraction                          | median | 0.738 | 0.689 | 0.536 | 0.408 | 0.497 | 0.821 | 0.492 |
|                                       | mean   | 0.083 | 0.053 | 0.020 | 0.021 | 0.022 | 0.061 | 0.026 |
| MISR BrS<br>AOD Fraction              | +/- σ  | 0.110 | 0.082 | 0.052 | 0.049 | 0.054 | 0.136 | 0.064 |
| AOD Fraction                          | median | 0.010 | 0.010 | 0.009 | 0.007 | 0.007 | 0.010 | 0.008 |
|                                       | mean   | 0.051 | 0.034 | 0.012 | 0.011 | 0.013 | 0.023 | 0.014 |
| MISR NonSph<br>AOD Fraction           | +/- σ  | 0.087 | 0.087 | 0.046 | 0.041 | 0.038 | 0.050 | 0.046 |
| AOD Fraction                          | median | 0.006 | 0.001 | 0.000 | 0.000 | 0.000 | 0.000 | 0.000 |
|                                       | mean   | 0.236 | 0.150 | 0.074 | 0.053 | 0.062 | 0.164 | 0.073 |
| MISR Very Small Particle AOD Fraction | +/- σ  | 0.146 | 0.151 | 0.124 | 0.112 | 0.126 | 0.166 | 0.130 |
| rarticle AOD Fraction                 | median | 0.241 | 0.107 | 0.000 | 0.000 | 0.000 | 0.133 | 0.000 |
|                                       | mean   | 0.582 | 0.521 | 0.521 | 0.488 | 0.481 | 0.646 | 0.502 |
| MISR Small Particle AOD Fraction      | +/- σ  | 0.215 | 0.247 | 0.270 | 0.249 | 0.231 | 0.177 | 0.245 |
| AOD Fraction                          | median | 0.611 | 0.581 | 0.544 | 0.506 | 0.517 | 0.665 | 0.533 |
|                                       | mean   | 0.000 | 0.056 | 0.162 | 0.253 | 0.256 | 0.061 | 0.224 |
| MISR Medium Particle<br>AOD Fraction  | +/- σ  | 0.000 | 0.169 | 0.258 | 0.295 | 0.286 | 0.138 | 0.284 |
| AOD Fraction                          | median | 0.000 | 0.000 | 0.000 | 0.125 | 0.154 | 0.000 | 0.061 |
|                                       | mean   | 0.094 | 0.080 | 0.040 | 0.041 | 0.046 | 0.045 | 0.046 |
| MISR Large Particle<br>AOD Fraction   | +/- σ  | 0.115 | 0.117 | 0.089 | 0.080 | 0.077 | 0.064 | 0.083 |
| AUD Fraction                          | median | 0.054 | 0.033 | 0.000 | 0.000 | 0.000 | 0.003 | 0.000 |
|                                       |        |       |       |       |       |       |       |       |

There is a need to differentiate between aerosol properties obtained *in situ*, from direct samples of particles, and those obtained from space-based remote sensing, e.g., by MISR, which represent an interpretation of column-effective radiance measurements. We therefore refer to RA particle size and light-absorption results as retrieved effective particle size (REPS) and retrieved effective particle absorption (REPA), respectively. These terms reflect the limitations of the retrieved quantities (i.e., qualitative, indirectly measured properties sampling the atmospheric column), while still communicating the measured content. For this and related studies, we observe along-plume changes in AOD, REPS, and REPA, combined with MINX-derived plume age estimates, to infer the relevant aging mechanisms for aerosol plumes. For example, Flower and Kahn (2020a,b) observed that decreasing AOD accompanied by decreasing REPS may indicate size-selective particle deposition, whereas constant or increasing AOD accompanied by decreasing REPS downwind may indicate the formation of secondary aerosols in volcanic plumes. In Canadian and Alaskan wildfire plumes, we inferred that plumes with relatively constant AOD yet decreasing REPA downwind probably experienced oxidation or particle hydration, whereas smoke plumes with decreasing AOD and decreasing REPS downwind are more strongly affected by dilution or gravitational settling (Junghenn Noyes et al., 2022).

# 2.2. Experiment Setting and Case Selection



True-color imagery and thermal anomalies in coincident data from the MODIS instrument aboard Terra, displayed in the NASA Worldview web application, were used to identify suitable fires within the MISR field of view. Well-defined plumes

with minimal cloud contamination and adequate optical thickness were favored for analysis. In Siberia, most plumes that are visible in the MISR imagery meet these criteria (unlike in tropical fire-prone regions such as the Amazon or southeast Asia). It is important to note that agricultural fires, although common in Siberia, are usually too small to be observed from space by MISR and MODIS and are therefore beyond the scope of the current study.

A total of 3,716 plumes spanning April through September across the years 2017-2021 were selected. Only plumes east of the Ural Mountains were considered -- the most westward plume was located at 60.25 °E, and the most southward plume was located at 48.69 °N (Fig. 1). The distribution of plumes by month and year can be found in Table 2 and Table S2 in the Supplement, respectively.



Figure 1. Locations of all fires included in this study (black dots), overlaid on (a) the 2017 MODIS IGBP land-cover-type map, and (b) the estimated percent of peatlands from Hugelius at al. (2021). See Table S3 in the Supplement for IGBP land cover type definitions. Note: "needle" = needleleaf; "broad" = broadleaf; "mosaic" = cropland/natural vegetation mosaics.

## 2.3. MODIS Fire Radiative Power and Land Cover Type

The MODIS/Terra Thermal Anomalies/Fire product (MOD14) was used to locate fire pixels and retrieve the 5 min FRP values at the time of MISR observation (Giglio and Justice, 2015). The user-defined MINX smoke-plume boundary, combined with the MODIS/Terra RGB imagery from NASA Worldview (https://worldview.earthdata.nasa.gov, last access; 29 October 2024) were used to assign each plume a mutually exclusive set of hotspots. The MOD14 product has a spatial resolution of 1 km, and it reports FRP based on a detection algorithm that evaluates differences in the hotspot vs. background brightness temperature using the 4 and 11 µm channels (Giglio et al., 2003). FRP is often used as a qualitative indicator of fire intensity; however, MODIS might underestimate FRP values under cloudy or dense-smoke conditions, when the active fire only partly fills the MODIS pixel, and for plumes in the smoldering phase that exhibit lower radiant emissivity and therefore produce lower FRP values (Kahn et al., 2008).

We systematically coupled the fire pixels with annual 0.5 km land cover type data from the MODIS MCD12Q1 product (Friedl and Sulla-Menashe, 2019), classifying the type(s) of vegetation burning in each hotspot using the International Geosphere–Biosphere Programme (IGBP) classification system. This product does not contain sufficient information to determine the actual *fuel* type consumed by fires, which also depends on factors such as meteorology and seasonality. However, land cover and fuel type are closely related, and we use the MODIS product to make inferences as to the types of fuels that are present. As the MCD12Q1 spatial resolution is finer than that of MOD14, some MODIS hotspots cover multiple land cover types, in which case we assigned land type as a split between the two that comprise the largest fractions of the fire pixel. Descriptions of the IGBP land cover types identified in this study are included in Table S3 in the Supplement.

## 250 2.4. Peatlands Distribution


Peat consists of plant detritus that has accumulated, typically on the forest floor, due to incomplete decomposition under water-saturated conditions, and is therefore a major reservoir of terrestrial carbon. However, climate-driven increases in peatland disturbance such as drought, thawing of permafrost, and higher fire frequency have the potential to reintroduce sequestered carbon into the atmosphere (Turetsky et al., 2015). As boreal forests are the major locations for peat fires compared to lower-latitude forest biomes, due to generally colder weather, wildfires in Siberian peatlands may have significant climate impacts in the future. For this study, we hypothesize which wildfires were burning in peatlands using GIS grids of peatland properties from Hugelius et al. (2021). The dataset provides several different peat maps quantifying the percentage of peat coverage, its depth in cm, and peat nitrogen and carbon storage at a 0.1° x 0.1° resolution for the full northern hemisphere north 23° N (Hugelius et al., 2020).

## 260 2.6. MERRA-2 Reanalysis

We retrieve the estimated height of the planetary boundary layer (PBLH) for each plume from the MERRA-2 reanalysis model (Bosilovich et al., 2016; Gelaro et al., 2017; Global Modeling and Assimilation Office, 2015b). The PBLH data are provided

Deleted: ed

Deleted: highly

at  $0.625^{\circ}$  longitude  $\times 0.5^{\circ}$  latitude spatial resolution and hourly temporal resolution, so we select the data point closest to the time and location of each fire plume origin. Note that, throughout this work, we alternate between using the phrases "above the PBL" and "in the free troposphere (FT)" for plumes that we estimate were injected above the MERRA-2-defined PBL. These terms have identical meaning and are used interchangeably.

Figure 2. Monthly variability in (a) the number of plumes observed, and (b) the percentage of plumes in the free troposphere. Each bar is divided by color according to the relative contribution from each fire type in the given month, with quantitative annotations where space allows. For example, 1600 plumes were identified in July (a), 7% of which were F fires, 20% of which were W fires, 36% of which were G fires, and 6.5% of which were P fires (with 31% of the plumes not falling into any of the four plume categories, i.e., "mixed" fires). Of the 1600 July fires, ~34% were in the free troposphere (b; <1% classified as F fires, 3.8% classified as W fires, 21.3% classified as G fires, 1.1% classified as P fires, and 7% mixed). Note that a plume is considered to be in the FT if its MINX-retrieved median height is 100 m greater than the PBL height as defined in the MERRA-2 dataset.





We calculate atmospheric stability profiles for the column above each plume using three-dimensional (3D) MERRA-2 meteorological data, reported every 6 h, at  $0.625^{\circ}$  longitude  $\times$   $0.05^{\circ}$  latitude spatial resolution (Global Modeling and Assimilation Office, 2015a). We define atmospheric stability as the vertical gradient of potential temperature, calculated using Eq. (1) (Holton, 1992), where S is the stability value at the midpoint between two model levels, d $\theta$  is the calculated difference in potential temperature between the levels, and dz is the difference in geopotential height. Potential temperature is calculated using Eq. (2) (Holton, 1992), where T and P are the atmospheric temperature and pressure, respectively, at altitude z. P<sub>0</sub> is the surface pressure (taken as 1000 mbar), R is the gas constant for dry air, and C<sub>p</sub> is the specific heat for dry air.

$$S = d\theta/dz \tag{1}$$

$$\theta = T(\frac{p_0}{p})^{R/Cp} \tag{2}$$

Formatted: Caption

We interpolate the temperature and pressure fields to the time of MISR observation for each plume,. The height of a stable atmospheric layer is defined as the height of the first maximum in the stability profile, so long as the stability is at least 1 K km<sup>-1</sup> larger than the layers above and below. This definition is consistent with how layers of relative stability have been defined in similar work by Kahn et al., (2007) and Val Martin et al., (2010), as well as in our study of Canadian and Alaskan wildfires.

To estimate the antecedent moisture conditions of vegetation and surface soil nearby a wildfire, we leverage surface moisture content from MERRA-2 hourly land surface diagnostics data (provided at 0.625° x 0.5° spatial resolution), averaged into daily means for the 24 hours preceding MISR observation of each plume. Surface soil moisture is provided on a scale of 0 to 1 (unitless), with 1 representing a fully saturated soil layer and 0 representing a dry surface.

#### 3. Results and Discussion




Figure 1 maps the locations of the plumes included in this study over Siberia, (a) superimposed on the 2017 MODIS IGBP land cover type map and (b) the Hugelius et al. (2021) peatlands distribution map. The largest number of plumes in our study was observed in 2019 (nearly 1,000), and the fewest in 2017 (529 plumes; Table S2 in the Supplement). Most plumes formed during the peak of the burning in season in July (1,600 plumes), followed by the month of August (1,165 plumes) (Table 2, Fig. 2a). Of the total number of thermal anomalies identified in the experiment, over 50% were located at least partially in savannas, ~40% in woody savannas, ~17% in open shrublands, ~17% in various forested biomes, and 

Deleted: Figure 2. Monthly variability in (a) the number of plumes observed, and (b) the percentage of plumes in the free troposphere. Each bar is divided by color according to the relative contribution from each fire type in the given month, with quantitative annotations where space allows. For example, 1600 plumes were identified in July (a), 7% of which were F fires, 20% of which were W fires, 36% of which were G fires, and 6.5% of which were P fires (with 31% of the plumes not falling into any of the four plume categories, i.e., "mixed" fires). Of the 1600 July fires, ~34% were in the free troposphere (b; 

Figure 5. MISR Ångström exponents (unfilled markers, dotted lines, left vertical axes) and mid-visible single-scattering albedos (filled markers, solid lines, right vertical axes) by smoke age for (a) Forest plumes, (b) Woody plumes, (c) Grassy plumes, and (d) Peat plumes, stratified by season. The points represent the mean values, and the whiskers show the standard deviations. In e), the MISR mid-visible AOD is plotted by age for the four plume types.

Deleted: ¶

## 3.3. Grassy Plumes

Grassy plumes make up the plurality of the plume types analyzed here. Median plume heights are statistically indistinguishable from those of both Forest and Woody plumes at ~1.4 km AGL. Plume heights increase from April to May, then decrease steadily to a minimum of ~0.86 km AGL in September, on average (Table S4C in the Supplement). Median-plume AOD for Grassy fires is, on average, statistically no different from that of Forest and Woody plumes at ~1.4, and exhibits the same seasonal trend as Forest and Woody fires when averaged across all time horizons. Smoke AOD as a function of smoke age in Grassy plumes is similar to the trends observed in Woody fires, with spring plumes exhibiting decreasing AOD downwind (although to a lesser degree than Woody plumes), and summer plumes maintaining relatively constant AOD values downwind (Fig. 5).

Grassy plume REPS is statistically no different from REPS in Forest and Woody plumes, with ANG ~1.68 on average (Table S4C in the Supplement). In April and May, REPS slightly decreases downwind (increasing ANG; Fig. 5), with plumes becoming increasingly dominated by small particles (Fig. 7). In summer months, REPS is relatively constant downwind as the fraction of medium-particle AOD never increases significantly, much less-so than is seen in Forest and Woody plumes, and the fraction of other particle sizes is constant after the first hour or two of aging.

Grassy plumes are darker than Woody and Forest plumes, with mid-visible SSA ~0.92 (p

Figure 6. MISR particle-type component AOD fractions (in terms of the contribution to the total AOD, from 0 to 1) by smoke age for (a) Forest plumes, (b) Woody plumes, (c) Grassy plumes, and (d) Peat plumes, stratified by season. BIS = black smoke; BrS = brown smoke; Nonabs = non-absorbing; Nonsph = non-spherical.

Formatted: Font: Bold, Font color: Auto

Deleted: 3

## 3.5. Summary and Inferred Aging Mechanisms

In summary, we find key similarities and differences in plume AOD, REPS, and REPA across the four plume types discussed here, with a significant seasonal component in many cases. Table 3 provides a summary of the main trends observed by season and plume type. Overall, summer plumes experience the most dramatic downwind changes in particle type, with plumes transitioning from BIS-dominated to non-absorbing-dominated. However, near-source fractions of BIS are much higher in spring and fall plumes, so summer plumes actually experience smaller decreases in plume REPA overall compared to other seasons. This suggests there may be seasonal differences in fuel properties, with spring and fall wildfires burning in areas that emit higher fractions of BIS compared to summer wildfires.

Fall plumes experience the most dramatic downwind changes in particle *size*, with REPS, as well as AOD, increasing as smoke ages. Together, this suggests that Fall plumes may be more subject to a combination of coagulation, hydration, and/or condensation compared to other seasons, as particles increase in size and therefore scattering more light.

Among the four plume types defined in this study. Peat plumes are the most distinct in terms of their overall particle properties and downwind evolution. Peat plumes are the optically thinnest overall (~0.99), with Forest, Woody, and Grassy plumes all exhibiting similar median-plume AOD values (1.25-1.4). This is likely driven by the fact that summer P plumes experience significant decreases in AOD with smoke age, whereas W and G plume AOD remains relatively constant and F plume AOD increases with age. (Trends in smoke AOD with age do not differ much between plume types in Spring and Fall.) Peat plumes are also the only plume type that experience decreasing REPS in summer months, whereas other plume types show more constant particle size downwind. Together, these suggest that dilution with background air may be more important in particle evolution for Peat plumes compared to other plume types, at least in the summer months when we observe the highest number of Peat plumes. As cleaner background air is mixed into the plume, this can shift the equilibrium for semivolatile compounds from the particle phase to the gas phase, resulting in stronger rates of evaporation and a particle size distribution that reflects smaller overall particles (May et al., 2013; Garofalo et a., 2019; Hodshire et al., 2019). In contrast, Forest, Woody, and Grassy plumes are likely more affected by a combination of oxidation, hydration, and/or condensation in the summer. This is evidenced by the transition from BIS-dominated smoke to non-light-absorbing-dominated smoke as plume age increases, together with the fact that AOD is constant or increasing. As particles oxidize, VOCs may condense onto their surface, which often reduces light-absorption and increases particle size. However, the fact that the overall retrieved ANG is relatively constant, despite the fuctuations MISR particle-size AOD fractions, suggests that the formation of small secondary aerosols may also be an important factor modulating REPS for these plumes.

The timescales over which the transition from BIS-dominated to non-absorbing dominated occurs varies between F, W, and G plumes in summer. Forest and Woody plumes experience the transition within the first 90 minutes, whereas Grassy plumes maintain a dominance of BIS until ~5 hrs. This may mean that plumes burning in areas associated with increased tree cover experience more rapid and significant oxidation or condensation/hydration compared to plumes burning in more open,

Deleted: f
Deleted: act that the
Deleted: fluctuate

grassy biomes, at least in summer. The increasing AOD with smoke age observed in Forest plumes supports this idea, as a potential combination of secondary aerosol formation and condensational growth could act to thicken smoke downwind.

Figure 6. Mean MISR particle-type component AOD fractions (in terms of the contribution to the total AOD, from 0 to 1) by smoke age for (a) Forest plumes, (b) Woody plumes, (c) Grassy plumes, and (d) Peat plumes, stratified by season. BIS = black smoke (black line and markers); BrS = brown smoke (red-brown); Nonabs = non-absorbing (pink); Nonsph = non-spherical (orange).

Figure 7. Mean MISR particle-size component AOD fractions (in terms of the contribution to the total AOD, from 0 to 1) by smoke age for (a) Forest plumes, (b) Woody plumes, (c) Grassy plumes, and (d) Peat plumes, stratified by season. BIS = black smoke (black line and markers); BrS = brown smoke (red-brown); Nonabs = non-absorbing (pink); Nonsph = non-spherical (orange).

# 3.6. Comparison with wildfires in Canada and Alaska

620

In previous work, we applied the same techniques used here to study the particle properties of 663 plumes throughout Canada and Alaska, spanning from May through September across the years 2016 to 2019 (Junghenn Noyes et al., 2022). Here, we

Formatted: Caption, Indent: First line: 0"

Formatted: Font color: Text 1

conduct a preliminary comparison of these results to Siberian plumes, with future work geared towards conducting additional in-depth comparisons as we investigate other major fire-prone regions and incorporate new datasets. The data products and processing methods described in this paper are identical to those used in the study of Canadian and Alaskan plumes, with the exception of the Hugelius et al. peatlands map, which was not leveraged in the previous study.

We find several distinct differences in smoke plume properties between Canada/Alaska and Siberia. In the former, the MISR particle properties are more strongly correlated with the dominant MODIS land cover type present, as we were able to stratify plumes using looser constraints and still observed statistically significant differences in many particle properties. This suggests that Canadian and Alaskan wildfire smoke properties are more dependent on fuel type than those in Siberia. For the purpose of this subsection, we re-define our Siberian wildfire plumes to match those used in our paper on Canadian and Alaskan plumes, with the addition of the Peat plume category, which was not incorporated into our previous study:

630

640

655

- "Forest-like" (FL) plumes, which contain any number of MODIS hotspots located in evergreen, deciduous, or mixed forests
- "Woody-like" (WL) plumes, which do not burn in forest but have at least 30% of their hotspots located in woody savanna, and up to 70% in savanna, grassland, or shrubland
- "Grassy-like" (GL) plumes, which also do not have forest but have at least 70% of their hotpots in savannas, grasslands, or shrublands, and no more than 30% in woody savanna
- "Peat" (P) plumes, defined using the same methods as in the rest of the text, as the peatland dataset extends to the entire northern Hemisphere

We compare how the observed wildfires are distributed among peatlands and the MODIS land cover types in an effort to understand how differences in particle properties between regions may be linked to differences in fuel types. Taken as an aggregate, about 15% of the MODIS hotspots were associated with forested land cover types for both regions. However, within the remaining 85%, Siberian plumes are heavily concentrated in grasslands/savannas, whereas Canadian/Alaskan plumes are more evenly partitioned between woody savannas and grasslands/savanna land cover types. The fraction of peat present in the observed plumes also differs between study regions — Canadian/Alaska plumes are associated with ~12.3% peat content on average, whereas, Siberian plumes are associated with ~7.6% peat (Figs. S4, S6 in the Supplement). The type of peat also differs; Canadian/Alaskan plumes are associated with relatively even fractions of non-permafrost and permafrost peatlands, whereas Siberian plumes strongly favor permafrost peatlands.

Although plume REPS was essentially equivalent, Siberian plumes had generally lower plume REPA compared Canadian/Alaskan plumes, with mid-visible SSAs of ~0.93 and 0.91, respectively. Although fires in both biomes contain ~50% BIS plume-wide on average, plumes in Canada/Alaska produce significantly higher fractions of BrS (6.0% vs. 2.6% of the total AOD) and somewhat lower fractions of non-absorbing particles (32% vs. 35%). This may be at least partially driven by differences in the type of vegetation burning, as discussed in the previous paragraph; peatlands tend to produce higher amounts of BrS compared to other fires, and the fraction of peatlands present was higher across all plume types in Canada/Alaska.

Deleted:

Formatted: Indent: Left: 0.25"

Deleted:

Deleted: with

Deleted: and

Differences in the meteorological conditions present at the time of MISR observation may also shape fuel availability and emissions. For example, coarse, woody fuels (known for emitting higher fractions of BrS compared to finer fuels; Urbanski, 665 2013) must be dry enough to burn. MERRA-2 reanalysis data suggests that Siberian plumes are consistently associated with higher mean surface moisture contents compared to plumes in Canada and Alaska, and that the difference is most notable in Peat plumes (70.4% vs. 62.2%; differences are on the order of ~3% for other plume types). Other factors such as temperature and wind speeds may also play a role; these will be investigated in future work.

Table 3. Qualitative summary of the key trends in MISR mid-visible AOD, REPS, and REPA by season

|      | Spring                                                                                                                                       | Summer                                                                                                                                                                                                                                                                                                                                                                                                        | Fall                                                                                                                                                                                                                                                                                                                                                                                        |
|------|----------------------------------------------------------------------------------------------------------------------------------------------|---------------------------------------------------------------------------------------------------------------------------------------------------------------------------------------------------------------------------------------------------------------------------------------------------------------------------------------------------------------------------------------------------------------|---------------------------------------------------------------------------------------------------------------------------------------------------------------------------------------------------------------------------------------------------------------------------------------------------------------------------------------------------------------------------------------------|
|      | Small particles dominate by far, but<br>there are significant fractions of<br>very small, medium, and large<br>particles throughout all ages | Small particles dominate at least near-<br>source; very small and large particle<br>fractions are negligible after ~1 hr                                                                                                                                                                                                                                                                                      | Small particles dominate by far;<br>contributions from very small and large<br>particles are significant for first several<br>hours                                                                                                                                                                                                                                                         |
| REPS | REPS is relatively constant or<br>slightly decreases with smoke age                                                                          | F/W Plumes: AOD fraction of medium particles become competitive with that of small particles after ~3-4 hrs, but overall ANG values are relatively constant  G plumes: AOD fraction of medium and small particles relatively constant, no real change in ANG  P plumes: AOD fraction of medium particles much lower from the start; small particle AOD fraction increases significantly downwind, as does ANG | REPS increases downwind  AOD fractions of very small and large particles are significant for first several hours, after which point medium particle AOD fraction increases at their expense; small particle AOD fraction either constant or decreasing  Most dramatic change in REPS is in P plumes, in which medium particles eventually become dominant over small particles after ~6 hrs |
|      | BIS dominates, but there are<br>significant fractions of BrS<br>(especially in F plumes) and some<br>non-spherical particles                 | Particles are a mixture of BIS and non-<br>absorbing particles; no significant<br>contribution from BrS or non-spherical<br>particles                                                                                                                                                                                                                                                                         | Particles are a mixture of BIS and non-<br>absorbing particles; no significant<br>contribution from BrS or non-spherical<br>particles (although more than in summer)                                                                                                                                                                                                                        |
|      |                                                                                                                                              | Near-source AOD fractions of BIS are<br>lower than in spring and fall                                                                                                                                                                                                                                                                                                                                         |                                                                                                                                                                                                                                                                                                                                                                                             |
| REPA | BIS remains dominant across all<br>smoke ages, but REPA does<br>decrease downwind                                                            | F/W plumes: BIS dominates near-<br>source until ~1-1.5 hrs, after which<br>point non-absorbing particles<br>dominate; REPA decreases  G plumes: BIS dominates until ~5 hrs,<br>after which point non-absorbing<br>particles dominate; REPA dramatically<br>decreases                                                                                                                                          | BIS is always dominant, but its AOD fraction does decrease downwind  REPA is very high near-source; this season experiences the most dramatic decreases in REPA with smoke age even though BIS is always dominant                                                                                                                                                                           |
|      |                                                                                                                                              | P plumes: BIS is always dominant;<br>REPA only slightly decreases                                                                                                                                                                                                                                                                                                                                             |                                                                                                                                                                                                                                                                                                                                                                                             |

| AOD | F/G plumes: AOD constant or<br>slightly decreases downwind<br>W/P Plumes: AOD decreases<br>downwind | F Plumes: AOD increases downwind<br>W/G plumes: AOD increases slightly<br>for first few hours, then stays relatively<br>constant | AOD increases downwind |
|-----|-----------------------------------------------------------------------------------------------------|----------------------------------------------------------------------------------------------------------------------------------|------------------------|
|     | Plumes are optically thinnest in spring                                                             | P Plumes: AOD decreases downwind                                                                                                 |                        |




Seasonal trends in median plume heights, as retrieved by MINX, are indistinguishable between Siberia and Canada/Alaska, with the exception that the Siberian wildfire season begins a month earlier, at least over the time periods included in these studies (Fig. S5a in the Supplement). The retrieved plume height values themselves are similar between the two regions, with maximum plume heights being statistically indistinguishable. Median plume heights are greater in Siberia compared to Canada/Alaska (1.35 km AGL vs. 1.22 km AGL, p<0.05); however, the regional difference is not large relative to the aggregated uncertainties in the MINX retrievals.

When averaged across all months, the height of the PBL is statistically indistinguishable across the two continents. Similarly, although exact PBLH values differed in a given month, overall monthly trends in PBLH are similar. The stability at the top of the PBL is also indistinguishable between the two regions, ~5 K/km on average. Of the plumes observed in both studies, Siberia wildfires injected into the FT at a rate 7% greater than that observed in Canada/Alaska. Median plume FRP is consistently lower in Siberia across all three observed seasons and across all four plume types (32.77 W/m² vs. 47.44 W/m², as a whole). Cumulative plume FRP (the sum of all hotspots in a given plume) is lower in Siberia across all three seasons, but the only statistically significant differences between plume types were observed in GL and P plumes (for, which values in Siberia were lower). The lower overall FRP in Siberia is consistent with current knowledge on how fires burn differently in the two continents - in Canada and Alaska, crown fires are more common, which are generally more intense than the surface fires that predominate in Siberia (Kharuk et al., 2021; Rogers et al., 2015). The observed differences in FRP, the higher plume heights in Siberia, and the greater rate of FT injection in Siberia all suggest that fire intensity may not play as strong of a role in plume-rise processes in Siberia.

In meteorological spring, Siberian plumes are optically thinner compared to Canadian/Alaskan plumes (0.871 vs. 1.82 mid-visible AOD), whereas the two regions share similar AOD values in summer and fall. Furthermore, the seasonal intracontinental trend in AOD differs between the two regions; plumes within Siberia are at their optically thinnest in spring, and plumes in Canada/Alaska are at their optically thickest in spring. Future work will aim to investigate the potential driving forces behind these differences, such as differences in meteorological seasonality.

The nature of downwind particle aging also differs between the two <u>regions</u>, Siberian plumes experience more rapid transitions from BIS-dominated to non-absorbing-dominated plumes. Furthermore, unlike in Canadian and Alaskan plumes, medium particles never truly dominate over small particles as smoke ages in Siberia.

Deleted:

Deleted: in

Deleted: continents



As has been mentioned, further work is needed to tease out some of the driving forces behind the observed differences in smoke plume properties between Siberia and Canada/Alaska. Although both regions are major boreal zones, it is well-known that different tree species dominate in each continent (Rogers et al., 2015), that the regional meteorology is also different, and therefore that the response to fires is likely to be different as well. Further investigation is certainly warranted. However, we note that the current study represents the first large-scale regional comparison of wildfires between the world's only two large, vegetated boreal zones, which together emit an average of 9.1% of global fire emissions (van der Werf et al., 2010).

4. Conclusions

This work, focused on Siberia, represents the second instalment in an ongoing effort to characterize wildfire smoke particles and smoke-plume evolution for the major burning regions across the globe. We apply known relationships between particle chemistry/microphysics and the optical signatures retrieved by MISR to create a regional inventory of particle size (four qualitative size bins ranging from very small to large) and particle type (black smoke vs. brown smoke vs. soil/dust vs. non-light-absorbing particles), as well as the inferred particle aging mechanisms present (e.g., dilution, oxidation, hydration, etc.) and the modulating forces behind these. The multi-angle nature of MISR observations allows us to derive plume-age horizons that provide us with the timescales over which critical particle type and size transitions occur under different conditions, representing new territory in wildfire smoke aerosol remote-sensing science and regional-scale statistical characterization.

Specifically, we find distinct patterns in smoke plume properties when the data are partitioned into four categories based on the relative fractions of forests (F plumes), woody savanna (W plumes), and savannas/grasslands (G plumes) present from the MODIS IGBP land cover type dataset, and peat (P plumes) present from a dataset provided by Hugelius et al. (2021). We find P plumes to be the most distinct of the four plume types, containing the smallest, darkest particles, and having the lowest plume AOD and lowest median plume heights. Forest and Woody plumes are the overall brightest plume types, whereas Grassy plumes exhibit REPA between that of F/W and P plumes. We observed no statistically significant differences in plume AOD, median plume heights, or REPS between F, W, and G plumes in Siberia.

We also find distinct seasonal differences in the retrieved particle properties, with plumes observed in spring (April and May) containing the highest overall AOD fractions of both BIS and BrS as well as the smallest particles, compared to other seasons. Spring plumes also exhibit the lowest overall plume AOD compared to summer and fall months. Plumes burning in spring and fall are associated with higher fractions of light-absorbing particles near the source, and therefore experience larger swings in REPA compared to plumes observed in summer, although they maintain their BIS-dominance throughout their observed lifetimes. In contrast, summer plumes experience a transition from BIS-dominated to non-light-absorbing-dominated. We also observe that the timescales over which particle type transitions occur differ significantly between the four plume types in summer, with non-light-absorbing particles becoming dominant over BIS in less than 2 hours in F and W

Deleted: Despite the fact that

plumes and ~6 hours in G plumes, but never in P plumes. Similarly, the fraction of medium particles increases downwind to be nearly equal to or slightly dominant over that of small particles after ~3 hours in F plumes, increases but never quite equals the small-particle AOD fractions in W plumes, and barely changes at all in G plumes. In P plumes, the AOD fraction of small particles increases downwind at the expense of other particle size categories, and plume AOD decreases. However, P plumes are the only plume type with ANG values that reflect a meaningfully changing particle size distribution. Based on these trends, we infer that the plume types experience varying types and degrees of atmospheric aging. Namely, we infer that:

- P plumes experience less oxidation and/or condensation/hydration compared to F, W, and G plumes, evidenced by
  the higher overall absorbing aerosol fraction retrieved by MISR
- Dilution may play a larger role in particle size evolution for P plumes, supported by decreasing particle sizes downwind combined with reduced AOD downwind
- 3. There may be seasonal differences in fuel properties and meteorology, and therefore emissions, as spring and fallplumes contain much higher fractions of BIS near-source compared to summer plumes.

Previous work on Canadian and Alaskan wildfires allowed us to create a regional inventory of particle type and evolution that was stratified by MODIS land cover type. In this study, we have expanded upon those methods with the incorporation of peatlands extent, a greatly expanded sample size, and the analysis of seasonal differences in particle properties. We compare observations between the two regions and find that, overall, Siberian plumes are associated with lower FRP, less light-absorption, and lower fractions of BrS. Such differences in particle properties are likely because fires in Siberia burned in areas associated with comparatively lower fractions of peatlands and woody savannas, instead favoring grasslands. Particle-type transitions are also more rapid in Siberia compared to Canada/Alaska. Lastly, although particle properties were associated with land cover type in both continents, this relationship was much stronger in Canadian/Alaskan plumes.

Future work will involve applying the MISR RA and associated tools to plumes in other major fire-prone regions, such as the western United States, Australia, and the Amazon. As we expand our repertoire, more in-depth comparisons between regions will become possible. The incorporation of new or unique datasets, such as fuel modelling, will allow us to gain new insights into the relationships between particle properties and fire properties. Such large-scale observational constraints on smoke particle type, evolutionary mechanisms, and the timescales over which they occur represent new territory and could greatly benefit climate and air quality modeling efforts.

Competing Interests: The authors declare no conflicts of interest.



Code/Data Availability: The MISR Research Aerosol (RA) algorithm is a proprietary product. MINX is available for public use and can be downloaded at <a href="https://github.com/nasa/MINX">https://github.com/nasa/MINX</a>. The RA and MINX results for individual plumes can be found at the NASA Langley Atmospheric Data Center (ASDC) Distributed Active Archive Center (DAAC): [final URL is TBD].

Formatted: Indent: Left: 0.25"

Formatted: Indent: Left: 0.25", Numbered + Level: 1 + Numbering Style: 1, 2, 3, ... + Start at: 1 + Alignment: Left + Aligned at: 0.5" + Indent at: 0.75"

Deleted:

Deleted: needed

Field Code Changed

**Author Contributions:** The project was first conceptualized by R.K, and the development and design of the methodology were a collaboration between R.K. and K.J.N. The RA algorithm used in this project was developed by R.K. and James A. Limbacher. K.J.N. developed the tools used to process, analyze, and visualize all data presented here. Formal analysis of the results was conducted by K.J.N and R.K., who together wrote and edited the original draft.

**Funding:** The work of K.T. Junghenn Noyes is supported by NASA's Atmospheric Composition Modeling and Analysis Program (ACMAP) under Richard Eckman. R. Kahn is supported by ACMAP and the NASA EOS Terra and MISR Projects.

Acknowledgements: The authors would like to thank James A. Limbacher for his role in the development and maintenance of the RA algorithm.

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

| Page 15: [1] Deleted<br>1:59:00 PM | Junghenn, Katherine T. (GSFC-613.0)[UNIV OF MARYLAND COLLEGE PARK] | 3/24/25 |
|------------------------------------|--------------------------------------------------------------------|---------|
| Page 16: [2] Deleted<br>5:38:00 PM | Junghenn, Katherine T. (GSFC-613.0)[UNIV OF MARYLAND COLLEGE PARK] | 3/25/25 |
| Page 20: [3] Deleted<br>4:54:00 PM | Junghenn, Katherine T. (GSFC-613.0)[UNIV OF MARYLAND COLLEGE PARK] | 3/25/25 |