# Peer review of "Siberian wildfire smoke observations from space-based multi-angle imaging: A multi-year regional analysis of smoke particle properties, their evolution, and comparisons with North American boreal fire plumes"

_EGUsphere, 2025_

## Author Response (AR1)

**Review 1**

Using MISR and MERRA2 products, the authors extend previous analyses of Canada and Alaska wildfires to include those in Siberia. This is a generally excellent paper. It's timely, insightful, and informative. But my recommendation to the editor is a straightforward rejection because of Code/Data Availability concerns (see my fuller comment below). If the authors address these concerns, I will be happy to take another look. I very much like the rest of the manuscript; if it weren't for the Code/Data Availability, I would've recommended it for publication as-is (comments below are non-blocking).

**Dear Reviewer 1,**

Thank you for the kind words about the quality of our paper. We have addressed your comments to the best of our ability, including the concern over Code/Data Availability. Below you can find a bulleted list of your comments and our responses.

• L29: I think the word "however" is likely the wrong word here

Upon inspection, we agree that the flow between the first two sentences of the Introduction doesn't sound right. Lines 29-30 now read as: "Wildfires are an integral component of the Earth system, influencing ecosystem processes across the globe. Although a certain degree of fire activity is natural and expected, the past two decades have been marked by a surge in large, uncontrolled fires that often take significant tolls on human society."

• L32: I think man-made may require qualification or even better a citation

We have added several citations there to: 1) support the link between climate change and fire frequency/severity, and 2) support the fact that the climate change we are currently experiencing is man-made

• L80: Still not really clear which study exactly examined the Canada and Alaska wildfires? Which one is it? Maybe just cite it here!

We now cite our paper in that sentence.

L86: And here!

Done.

• L137: is the geometric standard deviation designated as well?

Yes, this information is provided in Table S1 in the Supplement. We have edited the Table 1 caption header and footnote to provide clarity to the reader about where this information can be found.

• L140: citation (or link?) for may be needed for "publicly available"

We have inserted the link to the page where users can download MINX.

• L255: I didn't check, but I assume that's the same way you also defined it in the other study for the sake of consistency?

Yes, we this is now clarified in the text.

Figure 4: I found this figure hard to decipher (panels are too small, etc.). Consider improving. You can unify the titles (N=22, ...) and labels (Forest, Woody, ...) on top of the first row; you can use just on y-axis label (and include with it the a) median plume height, etc. info. Then, you may have enough space to showcase the figures themselves instead of the BIG words. See Figure 5, 6, 7 for inspiration:)

Certainly. We have modified Figure 4 (and Figure 3, since it had the same format) using your suggestions. The panels are now larger.

■ Table 3: I find tables in general hard to read. Is there a better way to showcase this datarich info?

Ideally, we would using shading to help draw reader's eyes to certain key places, but the journal does not allow for shading. It is indeed a lot of data, and upon reflection we decided to move it to the Supplement. The median plume heights and median plume AOD are illustrated in Figures 3 and 4 so readers can more easily visualize the monthly trends. The particle property statistics are displayed in various other figures, although they are displayed by meteorological season rather than month.

• Table 4: this is a productive usage of the table format. Note a minor typo "but it's AOD does" should be "but its AOD does" in the 4th row of the Fall column.

We have fixed this typo.

• L511: The analysis and data are consistent across datasets/studies, right? If so, I would assure the readers here by stating that.

Yes, the datasets and analysis methods are consistent between the two studies, and we've added some clarification.

• Section 3.6: I understand the authors plan to release more work with further and more indepth analyses, but I think this manuscript will benefit from further contextualization and/or speculation. My comment here is vague, but it simply an invitation for the authors to do more here if they think it is warranted.

The great thing about science is that there's always more to add! There is probably room to add a little bit more, but the paper is quite detailed and long already. Plus, comparisons between regions will be more interesting and fruitful once we've added even more biomes and datasets (in future papers).

Code/Data Availability: After reading this manuscript, I got pretty excited about potentially using the data and/or taking a look the amazing underlying dataset. In my opinion, this Code/Data Availability section is unacceptable and as such I don't think this work can be published without better disclosure of the underlying data AND some reproducibility code (to reproduce figures and tables in this manuscript). My request here, of course, does not apply to the noted propriety algorithm; you can keep that secret all you want. It applies to the raw data produced by this work, and especially the raw data used to make the scientific statements in this work. Also, please explicitly cite and point the reader to where they can find the "user-friendly" MINX. Anything short of full disclosure, the manuscript should be rejected and other reviewers shouldn't waste time on it. I understand the authors wrote "[final URL is TBD]" but I am sorry, let's not waste reviewers' time before it is "determined"

We are glad to hear that our data may be of use to you, and plan to have it archived on the Langley DAAC with its own URL before the paper is published. Uploading the data and creating the landing page is a process and takes a bit of time. Also, we wanted to make sure that there wouldn't be any changes to the dataset before archiving it. Now that we've received the reviewer comments and editor notes, we know that we have a complete dataset ready for archiving. We've also added the link for users to download MINX. And, FYI, the MISR RA is a research code – it is not designed nor supported to be run by general users. We know this would require a substantial effort for which we do not have any support. (One of us was part of the MINX development team, so we have some idea of what is involved in making a code useable for others.)

 Supplement: same comment regarding tables and figures as I made regarding Figure 4 and Table 3 above.

We have edited Figure S2 to increase the panel size and be consistent with the changes we made to Figures 3 and 4 in the main text.

**Review 2**

I think this is overall an excellent paper that provides interesting, useful, and sometimes surprising (and thus even more valuable) insights into the properties and ageing processes of Siberian biomass burning plumes. The findings, especially in the context of the broader goal of performing this analysis in different BB regions of the world (of which two – North America and Siberia have already been processed), have a great potential to contribute to better understanding and modeling of biomass burning aerosol emission processes. The goals stated in the introduction were reached by the work and well discussed in the conclusions.

Several items below would make the presented information clearer, in my opinion, but I did understand the message and the presentation well in its current form.

**Dear Reviewer 2,**

Thank you for the positive words about our paper. We have addressed your comments to the best of our ability. Below you can find a bulleted list of your comments and our responses.

■ L33 – I would have liked to see a reference to a study (or several) linking aspects of climate change to changes in fires to support this important claim.

We have added several citations there to: 1) support the link between climate change and fire frequency/severity, and 2) support the fact that the climate change we are currently experiencing is man-made.

■ L55 – the use of "flame front" brought up the question if all the fires considered here were actively burning. Seeing a significant number of peatland fires – were any of these smoldering fires? Is this important here? It may be not, so either clarify, or maybe use another term here that indicates the location/source/point of combustion, but not paints the image of flame front.

All the fires identified were associated with MODIS fire pixels and produced plumes that were thick enough to be characterized by MISR. In this introductory section I don't know if it's necessary to clarify that. However, your point is noted, so we have replaced "away from the flame front" with "downwind."

• L57 – "...and alter particle scattering as well as CCN efficiency, especially..." – you may want to add absorption after scattering here. You show that absorption profile changes with aging as well.

We have added "and absorption" after the word "scattering."

■ L59, also L52 and L68 – there are a lot of references in this list at the end of the paragraph, that are relevant to different parts of the paragraph. For example, Dalirian et al, 2018 is about CCN, so I would much prefer it to be mentioned as a stand-alone reference after CCN are mentioned in L57 than in a long list of other references about everything in the paragraph. This referring to works where they belong makes it a lot more convenient for the reader to get additional information on the point of interest right there, rather than going through the entire list.

Fair point. We have moved around some of the references in this area in a way that's consistent with your request, but in some places there is still a large grouping of references as they discuss a combination of the aging factors we discuss (condensation, SOA formation, hygroscopicity, etc.) and so it makes sense to have them together at the end.

■ L78 – In the sentence "In previous work..." do you mean Jungen Noyes at al., 2022? It is again cited in the end of the paragraph, so I was wondering all the way there, since line 78. Please move the reference to immediately after "In previous work..."

Done.

■ Table 2 – are the plumes above 2 km AGL a subset of Plumes above MERRA-2 PBL? I guessed that yes, but didn't find a confirmation to it in the text.

No, they are assessed separately. The PBL is sometimes below 2 km AGL, especially in the colder months.

• Section 2.2 – Is this plumes dataset available somewhere? If it is a part of the bigger dataset of digitized plumes, from which you selected a subset of 3,716, then it would be helpful to know. If this has been created for this study specifically then it looks like a valuable set for the scientific community. Please create a supplemental table or a link to a text or ncdf or any other format file with geographic coordinates and relevant information for each plume, like, height, length, age, FRP, vegetation type... - pretty much whatever relevant information was used to produce tables and figures in this paper but for each plume. It will likely be a big table, but valuable for scientific community, and also solidify reproducibility of your results, should anyone wish to do any complimentary studies.

The plume dataset was created specifically for this study, and we are in the process of getting the data archived online at the NASA Langley DAAC.

• Fig 3 and Fig 4 – labels on individual graphs are very small and difficult to read especially in printed version. I had to magnify the digital copy. Every little graph in this figure has

"N=" a few times above it. These are unnecessary and can be removed, just make a note in the table description that these digits are number of cases. In most graphs the numbers themselves could be moved inside the rectangle. These would free up some space to "stretch" the graph rectangles to make them more legible.

We have edited the figures to be more readable.

■ Fig. 5 – yellow axis labels are impossible to read in the printed version. Maybe, axes and labels can be black, while the data remains depicted in the color it is now. And, again, the labels are very small. There is no reason the graph can't take up the whole page. If you transpose rows and columns, and represent the same data as 3 columns with 5 rows, it should fit nicely onto a vertical letter page.

We have also edited this figure to be more readable.

• Fig 6 and 7 – same comment as for Fig. 5 about difficult to read small color labels on axes, especially in printed version.

And we have edited these figures as well.

---

## Referee Report (RR1)

**Review of**

"Siberian wildfire smoke observations from space-based multi-angle imaging: A multi-year regional analysis of smoke particle properties, their evolution, and comparisons with North American boreal fire plumes"

Katherine T. Junghenn Noyes and Ralph A. Kahn

I find this paper to be excellent and thoroughly enjoyed reading it. It provides substantial information about various wildfire plumes and offers valuable insights into the aging of plumes and their distinct characteristics. The comparison of the results in this paper with previous findings from Canada and Alaska enhances its value. After reviewing the comments of earlier reviewers, I noted that the authors have comprehensively addressed all feedback and suggestions. I believe the manuscript is ready for publication as it is. I have included a minor suggestion to add a reference to the manuscript as follows:

Line 147-150: Please add this reference for cloud wind and heights retrievals using MINX software:

O'Neill, N. T., Ranjbar, K., Ivănescu, L., Blanchard, Y., Sayedain, S. A., and AboEl-Fetouh, Y.: Remote-sensing detectability of airborne Arctic dust, Atmos. Chem. Phys., 25, 27-44, https://doi.org/10.5194/acp-25-27-2025, 2025.